# Pediatric Hypothermia: An Ambiguous Issue

**DOI:** 10.3390/ijerph182111484

**Published:** 2021-10-31

**Authors:** Dominique Singer

**Affiliations:** Division of Neonatology and Pediatric Critical Care Medicine, University Medical Center Eppendorf, 20246 Hamburg, Germany; dsinger@uke.de; Tel.: +49-(0)40-7410-52723

**Keywords:** thermoregulation, body size, neonates, infants, children, hypothermia, hibernation, adaptation, hypoxia, drowning

## Abstract

Hypothermia in pediatrics is mainly about small body size. The key thermal factor here is the large surface-to-volume ratio. Although small mammals, including human infants and children, are adapted to higher heat losses through their elevated metabolic rate and thermogenic capacity, they are still at risk of hypothermia because of a small regulatory range and an impending metabolic exhaustion. However, some small mammalian species (hibernators) use reduced metabolic rates and lowered body temperatures as adaptations to impaired energy supply. Similar to nature, hypothermia has contradictory effects in clinical pediatrics as well: In neonates, it is a serious risk factor affecting respiratory adaptation in term and developmental outcome in preterm infants. On the other hand, it is an important self-protective response to neonatal hypoxia and an evidence-based treatment option for asphyxiated babies. In children, hypothermia first enabled the surgical repair of congenital heart defects and promotes favorable outcome after ice water drowning. Yet, it is also a major threat in various prehospital and clinical settings and has no proven therapeutic benefit in pediatric critical care. All in all, pediatric hypothermia is an ambiguous issue whose harmful or beneficial effects strongly depend on the particular circumstances.

## 1. Introduction

While hypothermia in large and/or adult mammals usually means a clearly pathological deviation from normal homeostasis, the situation is less obvious in small and/or juvenile beings. For example, hibernators, which spend part of their lives at more or less reduced body temperatures, are usually found among the small or smallest mammals. This is probably because small mammals have the highest metabolic rates relative to their body mass and therefore benefit the most from cooling. In turn, this implies an increased tolerance to hypothermia, which is generally observed in small mammals, whether they hibernate or not, as an adaptation to the increased risk of heat loss. Thus, children tend to recover from lowered body temperatures more easily than adults, even though—or because—they are much more susceptible to hypothermia just for “geometric” reasons. The conflicting nature of hypothermia is particularly obvious in the neonatal period, where it is a critical risk factor for perinatal adaptation and, at the same time, an important self-protective mechanism against hypoxia.

To discuss the ambiguity of being cold in childhood, first, some definitions and basic principles are outlined in terms of their relevance to mammals of small body size. Then, a detailed review of the harmful and beneficial effects of hypothermia in both neonates and children, including the historical origins of its clinical use, is provided. Finally, the “pros and cons” of low body temperatures in pediatrics are summarized, and a brief outlook on potential future developments is given.

## 2. Basic Principles and Definitions

### 2.1. Pediatric vs. Adult Thermoregulation

One of the most common misconceptions about thermal balance in children is that their thermoregulation is somehow “immature” and therefore deficient. However, this is not the case, except for preterm neonates, who undoubtedly exhibit true immaturity. The basic thermoregulatory problem of infants and children is simply their small body size and correspondingly large surface-to-volume ratio, resulting in a high rate of heat loss to the environment relative to a small amount of heat stored in the body. Yet, even (term) neonates are adapted to these adverse conditions by an elevated basal metabolic rate (amounting to approximately 2.5 W·kg^−1^ compared to approximately 1 W·kg^−1^ in human adults) and by a steep increase in (non-shivering) thermogenesis (NST) in their brown adipose tissue (BAT) [1,2]. Although their maximal metabolic rate (summit metabolism) is higher, it is already reached at temperatures that are still felt to be comfortable by adults. This can mislead healthcare professionals into underestimating the invisible increase in O_2_ consumption rate required to maintain a seemingly normal or marginally subnormal body temperature. Thus, the problem with heat balance in children is not so much the immaturity of thermoregulation as the narrower regulatory range and the impending metabolic exhaustion [3,4].

### 2.2. Accidental vs. Induced Hypothermia

When the cold defense is overwhelmed or exhausted, a drop in body temperature occurs, which is referred to as accidental hypothermia in contrast to the induced hypothermia in cardiac surgery or neurocritical care. Whereas in the latter, a direct temperature-dependent decrease in metabolic rate (by a factor of 1/2.0–1/2.5 per 10 °C drop in body temperature) is achieved by pharmacological suppression of thermoregulation, the temperature decline in accidental hypothermia is preceded by a maximal (i.e., roughly five-fold) increase in heat production rate [5,6]. However, especially in children, the boundary between the two types of hypothermia may be blurred: For example, when caregivers keep pediatric critical care patients “a little cooler” in an attempt to spare their metabolism, this can actually result in unintended cold stress due to the hidden onset of thermogenesis. On the other hand, massive cold exposure, such as in drowning accidents of small children in ice-cold waters, may override the cold defense mechanisms so quickly that the metabolic reduction predominates. Moreover, as the brain does not participate in the thermoregulatory metabolic increase, even accidental cooling may exert a neuroprotective effect comparable to induced hypothermia.

### 2.3. Induced Hypothermia vs. Natural Hibernation

In the past, induced hypothermia was sometimes called “artificial hibernation”, by analogy with the cold-related metabolic reduction in hibernating mammals. In the meantime, however, it has been shown that the drop in body temperature in mammals during hibernation (or, more generally, “torpor”) is preceded, not followed, by an endogenous reduction in metabolic rate, and that their thermoregulation is not “turned off” but actively shifted to a lower set point [7,8,9]. Remarkably, the extent of temperature decrease appears to be determined by a specific minimal metabolic rate that hibernators share with the very largest mammals (Figure 1): the smaller an animal and the higher its specific basal metabolic rate, the lower its body temperature is allowed to drop—ranging from around 33 °C in hibernating black bears to near-zero temperatures in the smallest rodent hibernators. This natural regularity is of considerable importance in the context of pediatric hypothermia, as it is one possible explanation for the tendency toward higher hypothermia tolerance that has long been observed in juvenile compared with adult individuals as well as in small compared with large species.

### 2.4. Natural Hypometabolism vs. Energy Failure

The endogenous hypometabolism to be observed in mammalian hibernation differs not only from the cold-induced metabolic reduction in clinical hypothermia but also from the typical course of energy failure in mammals. In the case of deficient oxygen (O_2_) supply (hypoxia) or blood flow (ischemia), aerobic energy production is replaced by anaerobic glycolysis (lactate fermentation); however, this is not efficient enough to meet the high energetic demands of mammalian tissues. Therefore, the metabolic rate begins to passively decline (just as the bulb of a flashlight becomes darker when the batteries are running out) until irreversible damage occurs by falling below a lower maintenance metabolism. In contrast, mammalian hibernation is an example of active metabolic adaptation to predictable (e.g., seasonal) periods of adverse environmental conditions (just as the bulb of a flashlight is provisionally dimmed to extend battery life). Remarkably, similar adaptive responses are found in fetal and neonatal mammals to increase their tolerance to O_2_ and substrate undersupply in the potentially hazardous perinatal period: In addition to the redistribution of blood circulation (diving reflex), suppression of (non-shivering) thermogenesis is one of those self-protective responses, underscoring the ambiguous role of hypothermia in early life [18,19,20].

## 3. Neonatal Hypothermia

### 3.1. Hypothermia as a Risk Factor in Neonatal Transition

Unintentional heat loss is one of the greatest, if not *the* greatest, risk factor in neonatal transition. The typical symptom in term babies is respiratory distress (tachypnea, dyspnea, grunting, cyanosis), which reflects the high O_2_ demand of non-shivering thermogenesis that can hardly be met by the just aerated lungs. If this is correctly recognized and appropriately treated by rewarming the baby, the respiratory symptoms quickly resolve. However, if the baby remains exposed to cold, the sustained thermoregulatory response may lead to increasing metabolic acidosis, which further worsens O_2_ supply by impairing pulmonary perfusion. Thus, neonatal hypothermia can lead into a vicious cycle of hypoxia (Figure 2) and is one of the pathways to Persistent Pulmonary Hypertension of the Newborn (PPHN), which sometimes even affects the therapeutic use of cooling (cf. below) in post-asphyctic newborns [21,22]. To avoid neonatal hypothermia and its adverse effects, appropriate thermal care is mandatory already in the delivery room, combining minimization of heat loss with optimization of heat supply and including all physical modes of heat transfer (conduction, convection, radiation, and evaporation) [4,23,24].

### 3.2. Hypothermia as an Outcome Predictor in Preterm Neonates

In contrast to mature babies, preterm neonates exhibit a true insufficiency of thermoregulation due to their even smaller body size, their permeable skin, and the absence of insulating white and heat-producing brown adipose tissue. The fact that premature and/or hypoxic neonates show little if any defense against heat loss, combined with anecdotal reports that they were able to survive a state of “suspended animation” at very low body temperatures [25,26], had led previous neonatologists to simply allow them to cool in order to protect them (in a supposed analogy to “induced hypothermia”) from respiratory distress and/or oxygen lack [27,28]. This was the first attempt to use hypothermia in neonatology, long before therapeutic hypothermia was “reinvented” as a treatment option for post-asphyctic neonates about 30 years later (cf. below). The rapid departure from this approach was due to the seminal work of Silverman and Day [29,30], who demonstrated the adverse effects of hypothermia on mortality and morbidity in preterm infants, a fact that has been reconfirmed by a number of recent outcome studies [31,32,33]. Therefore, even greater efforts must be made in preterm than in term infants to prevent inadvertent cooling in the delivery room and beyond [4,23,24], even though current evidence does not completely rule out the possibility that “hypothermia may be a marker of illness and worse outcomes by association rather than by causality” [34].

Noteworthy for this purpose, kangaroo care, which was originally invented as a substitute for missing thermal care equipment in developing countries and has since gained worldwide acceptance [35,36], also shows surprisingly high efficacy [37,38]. Apparently, skin-to-skin contact is such a good heat exchanger that it is now explicitly recommended as an early and even “immediate” thermoprotective procedure in preterm neonates [39,40].

### 3.3. Hypothermia as a Self-Protective Mechanism in Perinatal Hypoxia

Notwithstanding its deleterious effects, it must not be overlooked that hypothermia has also a biological role in perinatal adaptation [18,19,20]. In clinical practice, this becomes apparent by the fact that the more depressed a term infant is after birth, the higher the susceptibility to hypothermia. The physiologic reason for this is that the brown adipose tissue is adversely affected by hypoxia and acidosis [41], impairing the newborn’s ability to produce heat through non-shivering thermogenesis. What may seem like a symptom of harm is part of a biological adaptation strategy increasing hypoxia tolerance in newborn mammals [42]. Animal experiments have shown that mammalian neonates respond to hypoxia by a suppression of thermogenesis resulting in a kind of “induced” hypothermia (Figure 3). Due to the absent thermoregulatory metabolic increase and the subsequent cold-induced metabolic reduction, O_2_ consumption is effectively decreased. In addition, hypothermia prolongs the duration of “autoresuscitative” gasps that support cardiac function during asphyxia, thereby further increasing the chance of survival [43,44,45].

From this point of view, it may be beneficial not to warm an asphyxiated neonate too much, especially since peripheral vasoconstriction increases the risk of burns from radiant heaters that primarily act on the body surface [47,48]. However, this is contradicted by the fact that upon restoration of O_2_ supply, a relatively rapid onset of thermogenesis may occur, which may then per se impair postnatal adaptation due to the concomitant increase in metabolic rate. Hence, an uncontrolled drop in body temperature should be avoided unless there is a clear indication for therapeutic hypothermia (cf. below) in the neonatal intensive care unit.

### 3.4. Hypothermia as a Treatment Option after Perinatal Asphyxia

In contrast to the spontaneous hypothermia of hypoxic neonates, therapeutic hypothermia, which has become the standard of care for asphyxiated neonates in recent years, is a “post-impact” intervention. The rationale here is to attenuate the “secondary energy failure” (reperfusion/oxygen radical injury leading to apoptosis) that follows the “primary energy failure” (hypoxic/ischemic injury leading to necrosis) after a short latency period and to prevent the damage from spreading to primarily unaffected tissues [49,50,51]. Although the safety and efficacy have been proven in several large multicenter trials [52], the method is unable to reverse the hypoxic–ischemic damage that occurred during the asphyctic episode and thus has an overall favorable but by far not fully satisfactory effect [53]. Moreover, in view of recent adult studies showing that hypothermia does not confer a significant benefit over a temperature management targeted at maintaining normothermia [54,55], as well as similar findings in older children (cf. below), one may wonder whether neonatology will remain virtually the sole domain of therapeutic hypothermia. This notion has gained further support from a brand-new study suggesting that hypothermic treatment may even do more harm than good, at least under circumstances such as those prevailing in low- and middle-income countries [56,57]. In any case, additional or alternative therapies are currently being sought that modulate the “tertiary” phase of hypoxic–ischemic brain injury so as to promote regenerative (rather than scarring) processes in the damaged neonatal brain [58,59,60].

## 4. Pediatric Hypothermia

### 4.1. Hypothermia Tolerance and Intolerance in Children

When surgery of congenital heart defects began in the late 1950s and early 1960s, it was induced hypothermia that played a crucial role, allowing the brain to be protected from O_2_ deprivation during brief periods of circulatory arrest [61,62]. With advances in cardiac surgery, before the technique of extracorporeal circulation became more widely adopted, the method was even extended to allow one-stage repair of complex defects in profound hypothermia [63,64]. The groundbreaking success of this approach was partly based on a higher apparent hypothermia tolerance of children as compared to adults. The empirical impression may be due in part to the fact that children are more easily cooled and rewarmed by superficial methods simply because of their larger surface-to-volume ratio. Moreover, it has long been known that the propensity for cardiac arrhythmias is substantially lower in young children, presumably also for geometric reasons (smaller ventricular muscle mass), and that their typical mode of circulatory arrest, especially for extracardiac (e.g., hypoxic) reasons, is a more or less long-lasting bradycardia leading to terminal asystole rather than an abrupt onset of ventricular fibrillation [65,66]. Finally, it may indeed be that children tolerate somewhat deeper body temperatures than adults due to the greater distance of their higher specific basal metabolic rate from a uniform lower metabolic limit (cf. Figure 1).

Although from a biological perspective, the increased tolerance may be interpreted as an adaptation to the higher risk of cooling, the clinical benefit is mainly relevant under conditions of pharmacologically induced hypothermia. When healthy children are lost in the wild and/or exposed to cold ambient temperatures in an inadequately protected state, they are at significantly increased risk of accidental hypothermia due to the tremendous heat losses and the metabolic exhaustion resulting from the vigorous yet futile thermoregulatory increase in heat production rate—with the rewarming success being largely dependent on the degree of hypoxia involved in the underlying accident [67,68,69].

### 4.2. Adverse Effects of Unintentional Hypothermia in Children

Accordingly, children are threatened by hypothermia in prehospital and clinical settings whenever environmental conditions are unfavorable or thermal protection measures are neglected due to an underestimation of the imminent heat loss [69,70,71]. In this context, hypothermia often reflects the severity of the injury, the duration of the exam, and/or the complexity of the surgical procedure, and for this reason alone is a predictor of adverse outcome. In addition, as in adults, hypothermia impairs blood clotting in children and increases the risk of bleeding. A drop in temperature by 5 °C suffices that one is practically dealing with a fully anticoagulated patient. The pitfall here is that coagulation tests determined in the laboratory at an analytical temperature of 37 °C often do not reflect the full extent of the disorder [72,73]. Moreover, the necessary rewarming procedure involves some risks for the patient, be it from the rewarming shock during superficial heating (with subsequent peripheral vasodilation) or from the abrupt resumption of thermoregulation with a sharp metabolic increase and a common temperature overshoot. Even though (unlike in adults) cardiac arrhythmias are not a primary concern (cf. above), efforts to return to normal body temperature present an avoidable challenge in the intensive care of severely injured or critically ill children. Therefore, the prevention of hypothermia should be given top priority in the prehospital and perioperative management.

### 4.3. Surviving Profound Accidental Hypothermia in Young Children

Despite the adverse effect of hypothermia in the vast majority of cases, there are numerous reports in the literature of young children who survived drowning accidents in ice-cold waters without major neurological sequelae [68,74,75]. This is usually explained by the protective effect of profound hypothermia. However, it may be the sequence of events rather than the degree of cooling itself that explains these types of “miracles”. When a young child falls into a frozen lake, there are four main points that may contribute to a favorable outcome [74]:Due to the huge temperature gradient and the large surface-to-volume ratio, thermoregulation is overwhelmed so rapidly that the child is virtually “shock-frozen” and enters the state of hypothermia without major metabolic disturbances.Somewhat similar to fetuses at birth [19,20], young children appear to exhibit a particularly pronounced diving response (consisting of apnea, bradycardia, and peripheral vasoconstriction), resulting in a more sparing use of remaining O_2_ stores [76,77].As a result of their aforementioned lower propensity to cardiac arrhythmias [65,66], small children are less likely to experience early ventricular fibrillation, even in cold water, than adult shipwreck victims who may suffer cardiac arrest (of whatever type) before a protective degree of hypothermia has been reached. A preserved heartbeat, albeit at slow pace, results in a residual tissue perfusion, delaying the progression of ischemia/acidosis and providing a supplementary “central” and thus more homogeneous cooling of the most jeopardized brain.The higher basal metabolic rate of small beings implies a larger distance to a minimal metabolic rate that may allow lower temperatures to be tolerated (cf. Figure 1).

As a whole, these physiological responses combine accelerated cooling with delayed hypoxia, ensuring that critical hypoxia does not occur until a deep, protective degree of cooling has been reached (Figure 4). This has some similarity to avalanche victims, whose survival is also known to strongly depend on the “right order” of (first) hypothermia and (second) suffocation (and not vice versa) [68,78].

Obviously, however, the outcome of ice-water drowning and other similarly “spectacular” hypothermic accidents in young children [75,79] does not only depend on the order of pathophysiological events but also on the determination and perseverance in resuscitation efforts. Case reports suggest that continuous resuscitation even under apparently hopeless conditions (“no one is dead until warm and dead”) is of vital importance [80,81].

Moreover, although superficial rewarming would often be possible in children owing to their lower propensity to cardiac arrhythmias [65,66,82], the use of extracorporeal circulation seems to be beneficial especially in rewarming from profound hypothermia because of superior hemodynamic stability and control [68,69,75,79,83,84,85,86].

By contrast, it remains questionable whether intermittent therapeutic hypothermia (cf. below), which is often established instead of immediate total rewarming to normothermia, plays a beneficial role [87]. According to current clinical experience, it is apparently capable of delaying the development of cerebral edema, although with the risk of its abrupt manifestation after definitive rewarming.

Finally, to avoid an overly optimistic notion of the effects of cold, it should be emphasized that such miraculous cases of surviving profound accidental hypothermia in young children remain the absolute exception. In the vast majority of drowned children, as in other trauma victims, low body temperatures reflect prolonged hypoxia and/or delayed return of spontaneous circulation (ROSC) and are thus associated with an adverse rather than a favorable outcome [85,88,89,90].

### 4.4. Hypothermia as a Treatment Option after Hypoxic Events in Children

Given the increased tolerance of children and by analogy to the evidence-based benefit in neonates [49,50,51,52], it would be tempting to use hypothermia as a therapeutic strategy after hypoxic–ischemic events or even traumatic impacts in pediatric critical care. However, the available evidence, both in out-of-hospital or in-hospital cardiac arrest [91,92] and in traumatic brain injury [93,94], suggests that there is no benefit compared with a temperature management targeted at maintaining normothermia and preventing fever [95,96], which often requires some degree of cooling in children anyway. The disappointing evidence is consistent with the above-mentioned findings in adults [54,55] and may, in children, be additionally due to the heterogeneity of critical care patients and the invasiveness of intensive care procedures counteracting the modest benefit of slightly reduced body temperatures. Thus, except in rare cases in which immediate rewarming might be more harmful than temporarily maintaining an already lowered body temperature, therapeutic cooling is not currently considered a routine procedure in pediatric critical care.

## 5. Conclusions

### 5.1. Propensity and Tolerance to Hypothermia in Children

In conclusion, it is important to understand that with the exception of premature neonates, the propensity of infants and children to cool down is not, or not mainly, due to an immaturity of their thermoregulatory system of any kind. On the contrary, (term) neonates, infants, and children are comparatively well adapted to their unfavorable surface-to-volume ratio by an elevated basal metabolic rate and a high thermogenic capacity. Thus, it is not as much the cold itself but the narrower regulatory range and the imminent metabolic exhaustion that pose the primary risk.

If thermoregulation is actually overwhelmed, then infants and children seem to tolerate low temperatures somewhat better than adults. This is mainly due to the lower risk of cardiac arrhythmias (probably due to the lower myocardial mass) and may be explained, at least from a comparative physiological point of view, by a higher metabolic rate and a correspondingly larger “distance” to a critical minimal metabolic level.

### 5.2. Peculiar Role of Neonates with Regard to Hypothermia

The ambiguity between risk and benefit particularly applies to neonates, since suppression of non-shivering thermogenesis with subsequent lowering of body temperature belongs to their self-protective responses to hypoxia. Thus, the tendency to cool even faster than normal not only reflects the compromised state of a newborn but also provides the newborn with additional resistance to O_2_ lack as part of a complex adaptive strategy.

However, this does not mean that hypothermia does not pose a risk to neonates: This is true for term babies, in whom the thermogenic response—more specifically, the huge increase in O_2_ consumption rate resulting from non-shivering thermogenesis in the brown adipose tissue—may cause respiratory maladaptation and, at worst, lead itself into a vicious cycle of hypoxia. This is also true for preterm infants who, because of their intrinsic immaturity, cannot defend themselves against a drop in body temperature and are threatened by secondary complications such as cerebral hemorrhage.

### 5.3. Ambiguity of Being Cold in Childhood Emergencies

The ambiguity of hypothermia also pertains to children. On the one hand, it reflects the severity of an injury or the complexity of a diagnostic or therapeutic procedure, and it poses an avoidable additional challenge to emergency or critical care management, particularly in terms of bleeding risk and cardiorespiratory instability resulting from the rewarming process.

On the other hand, it can be truly game-changing in those rare cases where an otherwise healthy child is so overwhelmed by the cold that hypothermia precedes hypoxia and thus provides a substantial protective effect—as in the ever-cited spectacular drowning accidents of small children in ice-cold waters. Thus, the role of hypothermia in children ranges from an ominous outcome predictor in the majority of trauma victims to a potentially life-saving ambient factor in singular drowning cases.

### 5.4. Current Limitations and Future Prospects of Therapeutic Hypothermia in Pediatrics

Favored by the relative tolerance of children to low body temperatures and by the ability to cool and rewarm them rather quickly, induced hypothermia had been extensively used before the advent of extracorporeal circulation to enable the surgical repair of congenital heart defects, thus exploiting its protective effect during hypoxia/ischemia. However, hypothermia as a neuroprotective treatment after a damaging event remains controversial. In contrast to asphyxiated neonates in whom it is still regarded as an evidence-based option, an unequivocal benefit of therapeutic hypothermia in infants or children has not yet been demonstrated after either traumatic or hypoxic impacts. For now, targeted temperature management as in adult critical care, i.e., maintenance of normothermia and strict avoidance of fever, is considered the method of choice to prevent the aggravation of pediatric brain injury.

Recently, new interest from space agencies has been expressed in “artificial hibernation”, which—if a torpor-like endogenous metabolic reduction with subsequent decrease in body temperature was somehow transferable to humans—could represent a veritable paradigm shift in organ preservation [97,98]. In view of the fact that spontaneous metabolic reduction in neonates and increased hypothermia tolerance in infants and children bear some similarities to natural torpor, this is an area of research that should not be dismissed a priori as mere “science fiction” [99,100] and may someday replace the current therapeutic approach of metabolic suppression enforced by external cooling.

## Figures and Tables

**Figure 1 ijerph-18-11484-f001:**
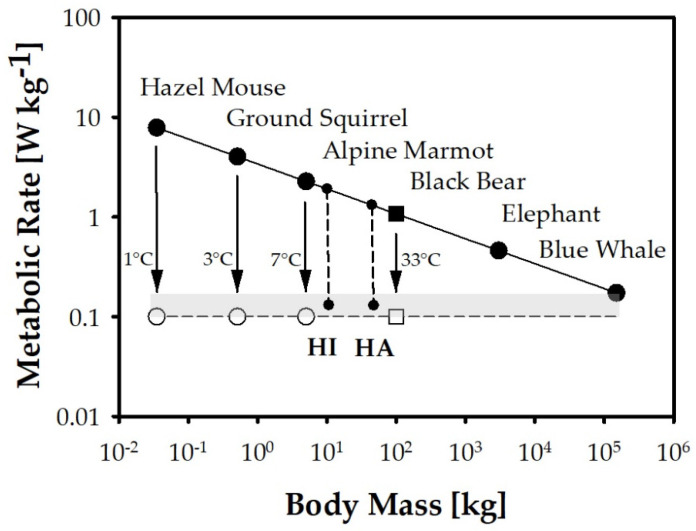
Tolerance to metabolic reduction as a function of body size. Following a general biological rule (“mouse-to elephant curve”), the specific (i.e., weight-related) basal metabolic rate of mammals (in watts per kg) decreases with increasing body mass [10,11,12]. Hibernating species (e.g., hazel mouse, ground squirrel, alpine marmot), while overwintering at near-zero body temperatures, reduce their metabolic rate to a fairly uniform minimal level that equals the specific basal metabolic rate achieved by the very largest mammals (elephant, blue whale) by body size alone [13,14,15]. Black bears fit into this overall hibernation pattern even though their body temperature remains higher (approximately 33 °C) and the relative amount of metabolic reduction is smaller due to their lower size-related basal metabolic rate [16,17]. Although there is no strict correlation between the (endogenous) metabolic reduction and the (concomitant) temperature decline in hibernators (both in hazel mice and in black bears, the magnitude of metabolic reduction exceeds the pure temperature effect), this relationship suggests an increasing hypothermia tolerance with decreasing body mass among mammals. This may also explain why human infants (HI) and small children tend to tolerate a higher degree of cold-induced metabolic reduction than human adults (HA), owing to their larger metabolic “drop height” [5,6].

**Figure 2 ijerph-18-11484-f002:**
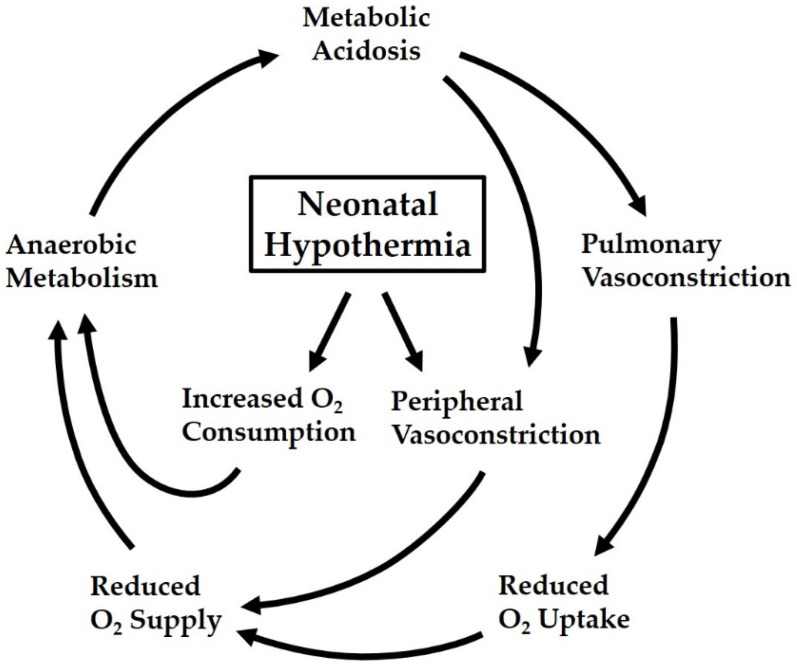
Vicious cycle of hypothermia-triggered hypoxia in neonates. The increase in O_2_ consumption rate caused by non-shivering thermogenesis in brown adipose tissue conflicts with the limited O_2_ supply by the just aerated lungs, especially under conditions of cold-induced peripheral vasoconstriction. The metabolic (lactate) acidosis resulting from higher anaerobic metabolism leads to an increase in pulmonary vascular resistance, which further impairs O_2_ uptake in the lungs and thus aggravates the mismatch between O_2_ demand and supply (adapted from [3]).

**Figure 3 ijerph-18-11484-f003:**
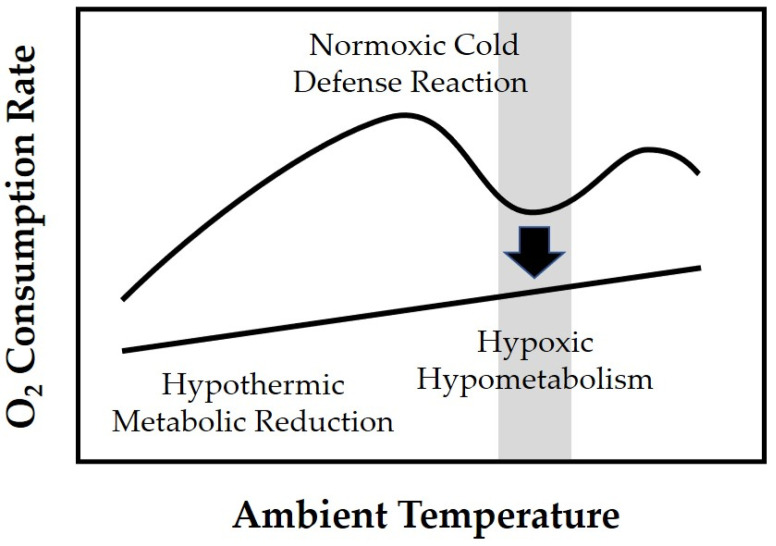
Self-protective role of hypothermia in hypoxic neonates. As hypoxia and acidosis exert a suppressive effect on non-shivering thermogenesis in the brown adipose tissue, the usual thermoregulatory increase in O_2_ consumption rate with decreasing ambient temperature is replaced by a direct cold-induced metabolic reduction, similar to induced hypothermia. Moreover, a spontaneous decrease in O_2_ consumption rate (arrow) in response to hypoxia may already occur at thermoneutral temperatures (shaded area), suggesting that neonates have a “hibernation-like” ability to reduce their metabolic rate (hypoxic hypometabolism) before any drop in body temperature (adapted from rat study data by Mortola [46]).

**Figure 4 ijerph-18-11484-f004:**
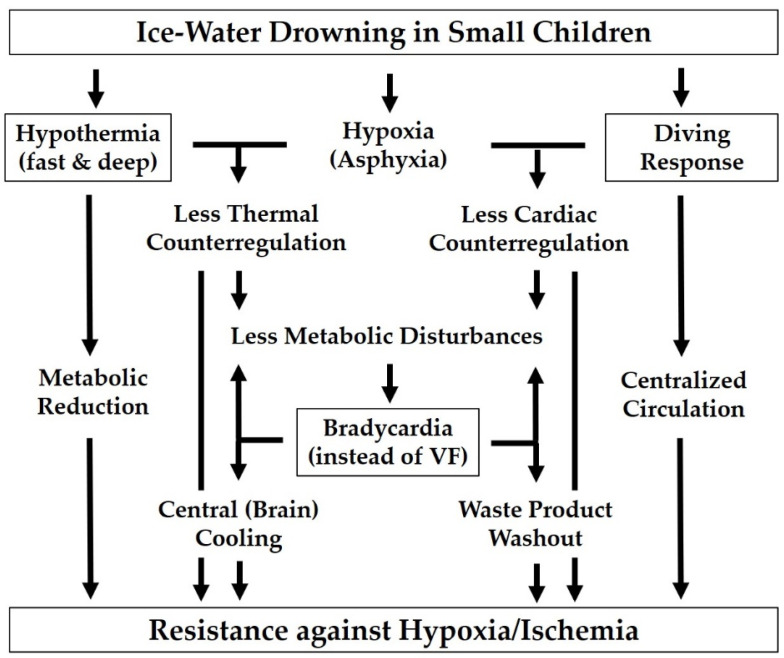
Factors affecting favorable outcome in drowning accidents of young children in ice-cold waters. Both the abrupt overwhelming of the cold defense reaction and the marked diving response attenuate the resulting metabolic disturbances, thereby enhancing the intrinsic resistance of the child’s heart to ventricular fibrillation (VF) or other types of cold-induced cardiac arrest. The sustained slow heart beat provides central brain cooling in addition to continuing washout of waste products from the tissues and thus prevents hypoxia/ischemia from assuming a critical level before a protective degree of hypothermia has been reached (adapted from [74]).

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
