# Peer review of "Pediatric Hypothermia: An Ambiguous Issue"

_ijerph, 2021, doi:10.3390/ijerph182111484_

Round 1

Reviewer 1 Report

The Author should be applauded for this extensive work. The manuscript describes in details various aspects of hypothermia in children. This seems to be very useful for clinicians.

However, some changes in the manuscript would be beneficial to readers.

General

In patophysiology of accidental hypothermia in children, information about hypothermic cardiac arrest (CA) is lacking. What is the most common heart rhythm of CA?; what is the temperature threshold of CA occurrence?; are there any differences between children and adults? Medical personnel of prehospital emergency services is familiar mainly with adult patients, hence such information would be very useful.

List of references contains non-english articles. It is a serious obstacle to further reading for readers worldwide. Suggest to replace them with english-language articles. Moreover, original studies rather than other reviews should be cited.

Specific

lines 55-59. "Although the maximum .... subnormal body temperature". This sentence is very long and unclear. Consider rephrasing.

l. 66. "decrease ... by a factor of 2 ..." Is it correct? An increase is described with a factor of >1, while a decrease with a factor of <1. If an increase is by a factor of 2, an analogous decrease is by a factor of 0.5.

l.70-74. "For example ... metabolic rate". this sentence is unclear. Suggest to rephrase.

l.142. "... replacement of unavoidable ones ..." Replacement with what? Suggest rephrasing whole the sentence.

l. 239, 241. Consider omitting quotation marks.

l.257. Serum lactate concentration is commonly elevated in severely hypothermic patients. However, this is not caused by rewarming. On the contrary, rewarming leads to the decrease in lactate level. Suggest to omit or rephrase.

l.261.  "unnecessary" is quite confusing here. Consider replacing with "unexpected" or rephrasing e.g. ... challenge that could be avoided...

l. 281. "ventricular fibrillation". See remarks to Fig.4

l.283. "The sustained heartbeat..." This sentence suggests that children do not experience cardiac arrest in drowning. If the Author meant a delayed CA (when compared to adults), please provide a reference (preferably original study in english).

Figure 4. Box "Bradycardia (instead of VF)" and line 302. VF is not the only primary mechanism of hypothermic cardiac arrest. Moreover, VF is not so common as it was deemed in the past. What about asystole and PEA? Does temperature threshold of CA onset in children differs from that in adults?

l. 315-320. In this paragraph, the Author stated that low body temperature is associated with fatal outcome. However, in the abstract (l. 16-17), the Author wrote "… hypothermia … promotes favorable outcome", and in main text  (l. 296-297) "continuous resuscitation even under apparently hopeless conditions ("no one is dead until warm and dead") is of vital importance". What does mean "fatal outcome" specifically? When analysing outcomes in accidental hypothermia, it is necessary to distinguish asphyxia-related from non-asphyxia-related incidents, and witnessed from unwitnessed cardiac arrest. Consider also comparing outcomes in normothermic and hypothermic cardiac arrest (especially when extracorporeal life suport was applied).

l. 326-327. "... targeted temperature management that maintains body temperature in the normal range ...". Targeted temperature management (TTM) is not a synonyme of normothermia. This includes maintaining normothermia as well as therapeutic hypothermia. Please rephrase to be more specific.

l. 365-373. Adding the paragraph about drowning into the section pertaining to injuries can lead readers to the false conclusion that hypothermia may be beneficial in trauma victims. Suggest to separate these two different items. Please consider also highlighting the impact of post-traumatic hypothermia on mortality.

Author Response

The Author should be applauded for this extensive work. The manuscript describes in details various aspects of hypothermia in children. This seems to be very useful for clinicians.

However, some changes in the manuscript would be beneficial to readers.

---> Thank you for the kind comment and useful advice which we have tried to implement as follows.

General

In patophysiology of accidental hypothermia in children, information about hypothermic cardiac arrest (CA) is lacking. What is the most common heart rhythm of CA?; what is the temperature threshold of CA occurrence?; are there any differences between children and adults? Medical personnel of prehospital emergency services is familiar mainly with adult patients, hence such information would be very useful.

---> A valid point indeed. Unfortunately, however, the published evidence on this topic is more than scarce, studies on cardiac arrest in (drowning) children (or infantile mammals, respectively) are literally non-existent. The only citable paper (#65) has now been supplemented by a second one (#66), and the mainly empirical (and partly hypothetical) considerations on the nature of cardiac arrest in young children have been described somewhat more explicitly than in the first version of the manuscript.

List of references contains non-english articles. It is a serious obstacle to further reading for readers worldwide. Suggest to replace them with english-language articles. Moreover, original studies rather than other reviews should be cited.

---> Thank you for the suggestion. Some of the German references are the only ones that can be cited as a source for a thought or a figure. Others were either removed or replaced by more or less equivalent references in English. Reviews were cited primarily when they provided additional information beyond the scope (space) of this paper (e.g., a comprehensive listing of previous case reports). Where this was not the case, and particularly if they were also written in German, they were removed or replaced accordingly.

Specific

lines 55-59. "Although the maximum .... subnormal body temperature". This sentence is very long and unclear. Consider rephrasing.

---> This is true. The „tapeworm“ sentence has now been split into two shorter and hopefully clearer ones.

l.66. "decrease ... by a factor of 2 ..." Is it correct? An increase is described with a factor of >1, while a decrease with a factor of <1. If an increase is by a factor of 2, an analogous decrease is by a factor of 0.5.

---> You are of course right. However, the so-called "Q10 value" (i.e. the factor by which the metabolism increases or decreases per 10° increase or decrease in body temperature) is usually given as 2.0 and 2.5, respectively. Therefore, I have now changed the numbers to 1/2.0 - 1/2.5.

l.70-74. "For example ... metabolic rate". this sentence is unclear. Suggest to rephrase.

---> Sure, the sentence has now been shortened and simplified.

l.142. "... replacement of unavoidable ones ..." Replacement with what? Suggest rephrasing whole the sentence.

---> That was rather abstract, indeed. A more common wording is now used instead.

l.239, 241. Consider omitting quotation marks.

---> Yes, of course, the quotation marks were removed from both „induced“ and „accidental“.

l.257. Serum lactate concentration is commonly elevated in severely hypothermic patients. However, this is not caused by rewarming. On the contrary, rewarming leads to the decrease in lactate level. Suggest to omit or rephrase.

---> This is exactly what was meant by the word "pre-existing". Since it seems to have been misleading anyway, I have decided to omit this point here.

l.261.  "unnecessary" is quite confusing here. Consider replacing with "unexpected" or rephrasing e.g. ... challenge that could be avoided...

---> Typical Germanism! „Unnecessary additional“ replaced by „avoidable“.

l.281. "ventricular fibrillation". See remarks to Fig.4

---> The sentence has been rephrased to avoid the unintended impression that ventricular fibrillation be the sole or major mechanism of cardiac arrest in adult shipwreck victims.

l.283. "The sustained heartbeat..." This sentence suggests that children do not experience cardiac arrest in drowning. If the Author meant a delayed CA (when compared to adults), please provide a reference (preferably original study in english).

---> Cf. above ("General"). The two citable references are now repeated here. Furthermore, the wording has been changed to emphasize the empirical or hypothetical character of this mechanism a bit more "honestly".

Figure 4. Box "Bradycardia (instead of VF)" and line 302. VF is not the only primary mechanism of hypothermic cardiac arrest. Moreover, VF is not so common as it was deemed in the past. What about asystole and PEA? Does temperature threshold of CA onset in children differs from that in adults?

---> Yes, but to emphasize the specificity of small children, I would prefer to leave the contrast with VF (which is often considered particularly typical of adult ice-water drowning) in the figure. To meet the objection, "… or other types of cold-induced cardiac arrest" was added in the legend.

l.315-320. In this paragraph, the Author stated that low body temperature is associated with fatal outcome. However, in the abstract (l. 16-17), the Author wrote "… hypothermia … promotes favorable outcome", and in main text (l. 296-297) "continuous resuscitation even under apparently hopeless conditions ("no one is dead until warm and dead") is of vital importance". What does mean "fatal outcome" specifically? When analysing outcomes in accidental hypothermia, it is necessary to distinguish asphyxia-related from non-asphyxia-related incidents, and witnessed from unwitnessed cardiac arrest. Consider also comparing outcomes in normothermic and hypothermic cardiac arrest (especially when extracorporeal life suport was applied).

---> Indeed, this was perhaps a bit too much of "ambiguity". Fortunately, most of the reviewer's comments had already been taken into account in the preceding paragraphs. The irritation raised here was probably due to a lack of specificity. The wording has therefore been changed so as to explicitly refer to the "vast majority of drowned children".

l.326-327. "... targeted temperature management that maintains body temperature in the normal range ...". Targeted temperature management (TTM) is not a synonyme of normothermia. This includes maintaining normothermia as well as therapeutic hypothermia. Please rephrase to be more specific.

---> Sure. The wording has now been changed so as to avoid the misleading impression that targeted temperature management refers to normothermia only.

l.365-373. Adding the paragraph about drowning into the section pertaining to injuries can lead readers to the false conclusion that hypothermia may be beneficial in trauma victims. Suggest to separate these two different items. Please consider also highlighting the impact of post-traumatic hypothermia on mortality.

---> The term „injured“ was erroneously used and has been removed. Also, an extra sentence has been added to clarify what exactly is meant by "ambiguity".

Reviewer 2 Report

The authors review the problem of the hypothermia in pediatric. The paper is well written and it is very informative and concise review of current literature. Figures and diagrams  are clear and English is good. 

I have to be congratulated with the authors. I have only three minor comments:

  1. I think an interesting and informative paragraph for the reader would be: The history of hypothermia in pediatrics.
  2. It would be interesting if the authors could add a future perspectives paragraph describing the direction towards which the scientific and clinical focus should be directed.
  3. Please review the references as more than 67% of them are older than 5 years.

Author Response

The authors review the problem of the hypothermia in pediatric. The paper is well written and it is very informative and concise review of current literature. Figures and diagrams are clear and English is good. 

I have to be congratulated with the authors. I have only three minor comments:

---> Thank you for your nice comment and helpful advice which we have tried to implement as follows.

1. I think an interesting and informative paragraph for the reader would be: The history of hypothermia in pediatrics.

---> Interesting suggestion, thank you. However, although the paper has a rather pathophysiological and clinical focus, it already contains quite a number of (more than usual) references to the "origins of hypothermia" in early neonatology and infant heart surgery. This is also one of the reasons why several older (milestone) papers are cited (cf. comment #3). In order to emphasize this „historical“ component a bit more, it has now been explicitly announced in the introductory remarks.

2. It would be interesting if the authors could add a future perspectives paragraph describing the direction towards which the scientific and clinical focus should be directed.

---> Valid point, thank you very much. Although probably not specific to pediatric hypothermia, a paragraph on "future perspectives" (a field in which we are involved ourselves, by the way!) has now been included in the paper’s concluding remarks.

3. Please review the references as more than 67% of them are older than 5 years.

----> You are absolutely right. On the one hand, this is due to the fact that some historical sources are explicitely referred to (see comment #1). On the other hand, and more importantly, it is because certain (pathophysiological) information is very difficult to find and has also gone somewhat "out of fashion" recently. Nevertheless, the bibliography has now been revised to ensure that only those earlier papers are cited that are "truly irreplaceable". In addition, a number of more recent papers/studies have been added to shift the "balance" a bit more towards the present.

Round 2

Reviewer 1 Report

The Author responded satisfactorily to my comments. Thank you for the opportunity of reviewing this manuscript.